# ciRS-7 expression is epigenetically regulated in cancer cells across human adenocarcinomas

Thea P. Paasch[1], Morten T. Jarlstad Olesen[1¤], Juan L. García-Rodríguez[1],
Adrienne M. Assmus[1], Robert A. Fenton[1], Jørgen Kjems[2,3], Henrik Hager[4,5],
Lasse S. Kristensen[1]*

1 Department of Biomedicine, Aarhus University, Aarhus, Denmark, 2 Department of Molecular Biology and Genetics (MBG), Aarhus University, Aarhus, Denmark, 3 Interdisciplinary Nanoscience Center (iNANO), Aarhus University, Aarhus, Denmark, 4 Department of Pathology, Aarhus University Hospital, Aarhus, Denmark, 5 Department of Molecular Medicine, Aarhus University Hospital, Aarhus, Denmark

¤ Current address: Karolinska Institutet, Solna, Stockholm, Sweden
* lasse@biomed.au.dk

## Abstract

Circular RNAs (circRNAs) constitute a large class of non-coding RNAs with gene regulatory capabilities, mainly through microRNA binding, a mechanism that has been linked to cancer development. The circRNA ciRS-7 (also known as CDR1as) is an interesting candidate as it harbors over 60 binding sites for miR-7, which is known to have tumor-suppressing properties. Here, we investigated the spatial expression patterns and epigenetic regulation of ciRS-7 across nine different adenocarcinomas originating from the colon, pancreas, ovary, endometrium, breast, stomach, bile duct, lung, and prostate. The study included primary patient samples and 18 different cell lines. ciRS-7 expression was analyzed using Reverse Transcription-quantitative PCR (RT-qPCR), single molecule *in situ* hybridization, and Nanostring nCounter, while epigenetic modifications were examined through bisulfite sequencing, Sensitive Melting Analysis after Real Time – Methylation Specific PCR (SMART-MSP), and chromatin immunoprecipitation. The functional relevance of epigenetic modifications was examined using DNA methyltransferase and histone deacetylase inhibitors. Across all adenocarcinomas, ciRS-7 was absent in the cancer cells in most of the primary tumor specimens, except for the breast tumors, while being expressed in the tumor microenvironment (TME). In line with this, ciRS-7 was not detected in most of the cell lines. Moreover, we demonstrated that DNA methylation and H3K9 acetylation, but not H3K27 methylation, are important epigenetic modifications that impact ciRS-7 expression. In conclusion, our data show that ciRS-7 is mainly expressed in the TME and is regulated through DNA methylation and histone acetylation across all major types of adenocarcinomas.

which permits unrestricted use, distribution, and reproduction in any medium, provided the original author and source are credited.

**Data availability statement:** All relevant data are within the manuscript and its Supporting Information files.

**Funding:** This study was supported by the Lundbeck Foundation (R307-2018-3433 to LSK), the Danish Cancer Society (R304-A17698 to LSK; R269-A15768 to JK), the Health Research Foundation of Central Denmark Region (A5377 to HH), the Independent Research Fund Denmark (10.46540/3103-00078B to LSK), the Carlsberg Foundation (CF20-013 and CF21-0243 to RAF), and the Novo Nordisk Foundation (NNF21OC0067647 to RAF). The funders had no role in study design, data collection and analysis, decision to publish, or preparation of the manuscript.

**Competing interests:** The authors have declared that no competing interests exist.

## Author summary

In cancer, the most extensively studied circular RNA to date, ciRS-7, has been proposed as an oncogenic driver. However, we show that ciRS-7 is absent in cancer cells across most major cancer types of glandular origin, i.e., adenocarcinomas. Interestingly, the genomic region controlling ciRS-7 expression possesses characteristics indicative of epigenetic regulation, but it has not been studied how these features affect ciRS-7 expression. Using both cell lines and primary patient samples, we investigated how the expression of ciRS-7 is regulated epigenetically across nine different types of adenocarcinomas. We discovered that ciRS-7 is completely absent in the cancer cells of most primary patient samples, while being abundant in the surrounding tumor microenvironment. In line with this, we found that ciRS-7 expression is lacking in most adenocarcinoma cell lines and is influenced by both promoter methylation and histone acetylation. This provides key insights into the epigenetic regulation of ciRS-7 in cancer, which is essential for guiding future research. In particular, artificial expression in cancer cells should be avoided, as such models may lack physiological relevance in the context of ciRS-7. Rather, research should focus on deciphering its oncogenic potential, mediated by cells in the tumor microenvironment.

## Introduction

Adenocarcinomas are the most common type of cancer, and are characterized by neoplastic growth of epithelial cells in glandular tissues [1,2]. Given the important role of gene regulation in cancer development, circular RNAs (circRNAs), a large class of non-coding RNA, have emerged as potential cancer drivers due to their gene regulatory capabilities [3–5]. CircRNAs are endogenously expressed, covalently closed molecules formed through an alternative splicing event that links a downstream splice-donor site to an upstream splice acceptor site, resulting in a so-called backsplicing-junction (BSJ) [6–10]. Most circRNA host genes are protein coding and, although many circRNAs are very lowly expressed and may arise as a result of aberrant splicing [11], some have been reported to serve as gene regulators through binding of microRNAs (miRNAs) or proteins [4,12,13]. The binding of miRNAs is denoted as "miRNA sponging" and is exerted by the circRNAs through specific miRNA binding sites, also known as miRNA response elements (MREs) [7]. However, despite this being proposed as the primary mechanism of action for the majority of circRNAs, only a few circRNAs actually contain more MREs than would be expected by chance [14]. Furthermore, the low expression levels of most circRNAs raise questions about whether the stoichiometry between the circRNAs and the miRNA target genes allows for efficient sponging in many cases [15].

An exception to this general rule is ciRS-7 (also known as CDR1as), which harbors numerous MREs for miR-7 and multiple other miRNAs [4,5,16], and is highly expressed in several tissues as well as in solid tumors [14,17]. ciRS-7 expression

is driven by one or more promoter regions associated with three linear transcripts, denoted as T1, T2, and T3, of the lincRNA, *LINC00632*, located on the X-chromosome [16,18]. A correlation between the expression of ciRS-7 and T3 has been found in the colon, while the T1 transcript correlates with ciRS-7 expression in the brain [18–20]. Furthermore, high ciRS-7 expression is associated with poor prognosis in multiple cancers, suggesting that ciRS-7 may function as an onco-gene [17,19,21–23]. It was initially believed to exert its oncogenic function by sponging of the tumor suppressor miRNA, miR-7, within cancer cells [3,18,22]. However, recent findings in colon and pancreatic cancer show that ciRS-7 is absent in the cancer cells but abundant in the tumor microenvironment [19,24,25].

Gene expression can be regulated in several ways, for example through epigenetic regulation by histone modifications and DNA methylation of promoter regions [26,27]. Hypermethylation of CpG rich regions, known as CpG islands, is asso-ciated with repression of gene expression, while post-translational modifications of histones can lead to both increased and decreased gene expression. Histone acetylation is associated with increased gene expression [28], whereas histone methylation, including H3K27me3, is generally considered a genetic repressor [29,30].

Here, we explored the expression and epigenetic regulation of ciRS-7 across various adenocarcinomas, including colon, pancreatic, ovarian, endometrial, breast, stomach, bile duct, lung, and prostate cancer. Our analyses involved both cell lines and laser-capture microdissected primary tumors, as well as full tissue sections. Expression was analyzed using single molecule *in situ* hybridization, Reverse Transcription-quantitative PCR (RT-qPCR), and NanoString nCounter, whereas epigenetic patterns were explored using bisulfite sequencing, Sensitive Melting Analysis after Real Time - Meth-ylation Specific PCR (SMART-MSP), and chromatin immunoprecipitation (ChIP)-qPCR. In addition, we examined how the expression of ciRS-7 was affected by treatment with the demethylating agent, 5-azacytidine (5-aza) and the histone deacetylase inhibitor (HDACi) Vorinostat (SAHA) in cell lines.

## Results

### ciRS-7 is not expressed in cancer cells within most primary tissues from nine different adenocarcinomas

Having previously demonstrated the absence of ciRS-7 in cancer cells in primary colon adenocarcinomas [19,24], we aimed to assess its prevalence across other types of adenocarcinomas. To this end we used chromogenic and fluorescent *in situ* hybridization and found that ciRS-7 was absent in the cancer cells, but expressed in the stroma, in most cases; 100% of colon, endometrium, ovary, and stomach tumors, 80% of pancreatic, prostate and bile duct tumors, 60% of lung tumors and 20% of breast tumors (Fig 1A-1I and S1 Fig). Images of all patient samples can be found in S1 File.

### Absence of ciRS-7 observed in most adenocarcinoma cell lines

Based on our initial findings, we wanted to explore how the expression of ciRS-7 in primary tissue samples was reflected in human cell lines. To this end, we analyzed 18 cell lines representing nine different adenocarcinomas, as well as a normal colon fibroblast cell line (Fig 2A), using RT-qPCR. ciRS-7 expression was generally absent (Fig 2B), except for the lung cancer cell line A549, which exhibited very high ciRS-7 expression, while the other lung adenocarcinoma cell lines, H1650 and H1975, and the endometrial adenocarcinoma cell line, KLE, exhibited low levels of expression. Additionally, it was observed that ciRS-7 was expressed in the colon fibroblast cell line CCD18-Co, as in line with previous findings [24]. T3 expression varied between cell lines (Fig 2C) and was significantly correlated with ciRS-7 expression (Fig 2D), whereas the T1 and T2 transcripts were undetected in all cell lines. Of note, significant correlation between ciRS-7 and T3 expression was also observed when removing the data point with the highest ciRS-7 and T3 expression ($R^2 = 0.49$, F-test: $P < 0.005$). The RT-qPCR results were validated using NanoString nCounter (Fig 2E-2G), which also showed that ciRS-7 expression significantly correlates with T3 expression (Fig 2G). The two methods were generally in good agreement, how-ever, using the nCounter technology, the H1975 lung cancer cell line exhibited high ciRS-7 expression (Fig 2E). This was in line with the T3 transcript also being detected at relatively high levels in this cell line using RT-qPCR (Fig 2C). Contrary

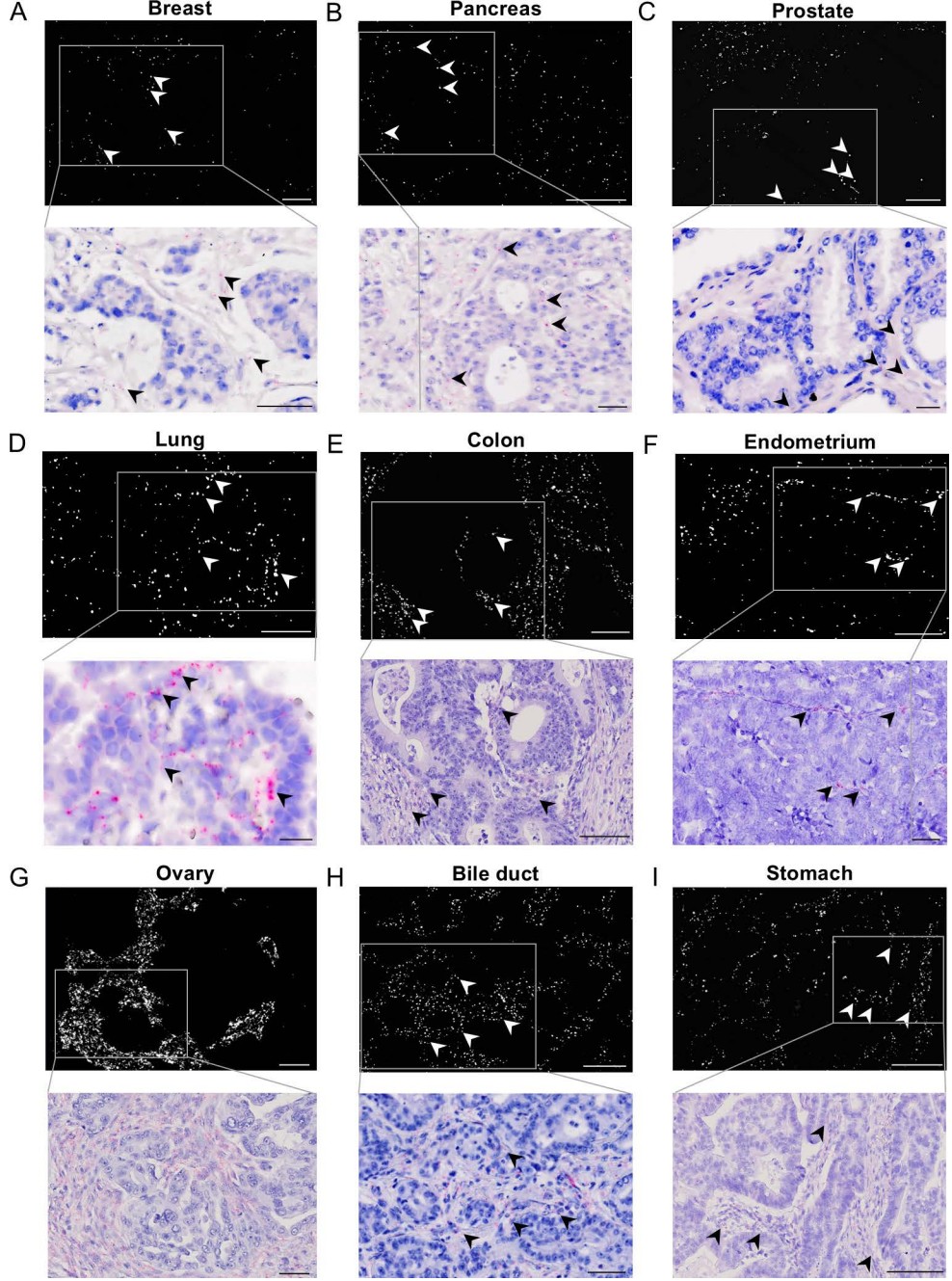

**Fig 1. ciRS-7 expression is absent in cancer cells from most primary adenocarcinoma tissue samples.** A-E) Chromogenic and fluorescent *in situ* hybridization of ciRS-7 in ductal breast (A), pancreatic (B), prostate (C), lung (D), colon (E), endometrial (F), serous ovarian (G), bile duct (H) and stomach (I) adenocarcinomas. In fluorescent (top) and brightfield (bottom) microscopy, the ciRS-7 signals are observed as white or pink dots, respectively. Arrows indicate selected florescent signals and their corresponding brightfield-images. Scale bars are indicated in the lower-right corners. Scale bar in fluorescent microscopy images: A = 50 μm, B = 100 μm, C = 50 μm, D = 50 μm, E = 100 μm, F = 100 μm, G = 200 μm, H = 100 μM, I = 200 μm. Scale bar in brightfield microscopy images: A-I = 50 μm.

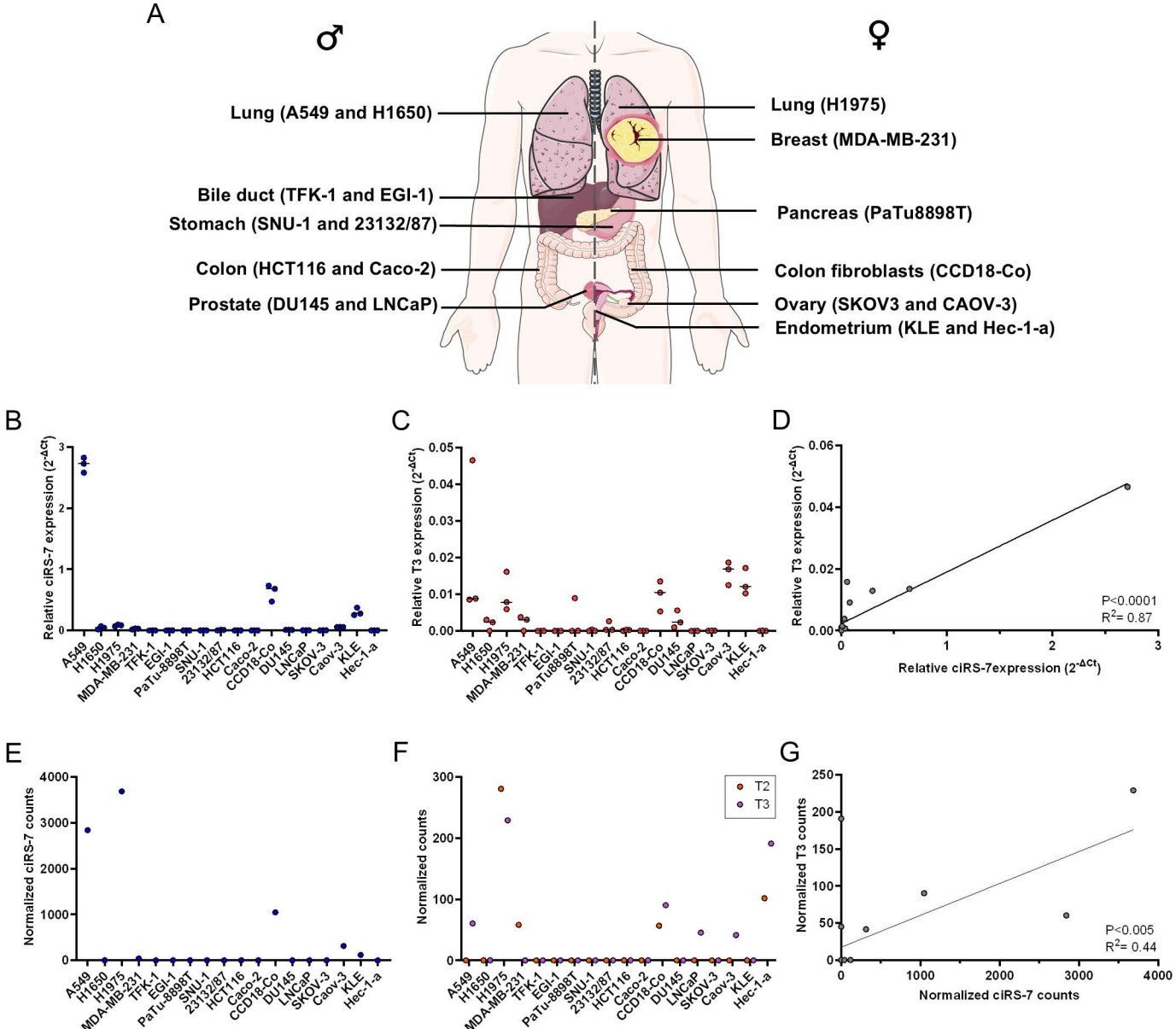

**Fig 2. Analyses of ciRS-7 and T3 expression across cell lines.** (A) Schematic representation of the biological origin and gender of the cell lines used in this study (created using Servier Medical art). (B-C) Relative expression of ciRS-7 (B) and T3 (C) in the cell lines measured by Reverse Transcription-quantitative PCR (RT-qPCR) as technical triplicates at the level of qPCR. (D) Scatterplot showing the mean relative expression of ciRS-7 (x-axis) and T3 (y-axis) in each cell line respectively. (E-F) ciRS-7 (E) and T2 and T3 (F) expression levels across cell lines measured with the NanoString nCounter technology. (G) Scatterplot showing the normalized ciRS-7 (x-axis) and T3 (y-axis) expression levels measured with the NanoString nCounter technology. For scatterplots in (D and G), linear regression statistics (F test) and Pearson's correlation coefficient (r) were employed to observe if the slope was significantly non-zero. Fig 2A was partly generated using Servier Medical Art, provided by Servier, licensed under a Creative Commons Attribution 4.0 Unported License. To view a copy of this license, visit https://creativecommons.org/licenses/by/4.0/.

to the RT-qPCR data where neither the T1 nor the T2 transcripts were observed, low T2 expression was observed using the nCounter technology in the CCD18-Co, Hec-1-a, A549, CAOV-3, and MDA-MB-231 cell lines (Fig 2F). To summarize, ciRS-7 and T3 expression correlated strongly, although expression was not observed in most of the cell lines.

## Absence of ciRS-7 expression in cell lines correlates with epigenetic silencing

Given the presence of a CpG island in the T3 promoter, our next aim was to examine whether ciRS-7 is regulated by DNA methylation. For this purpose, we used direct bisulfite sequencing [31]. Quantification of the methylation percentage for each individual CpG site was performed based on the peak intensity values in the electropherograms. It was observed that methylation levels varied between the cell lines, and that the cell lines exhibiting ciRS-7 expression were unmethylated for all the interrogated CpG sites (Fig 3A, 3B). Of note, we also observed that several female cell lines were completely unmethylated, indicating that the inactive X-chromosome is not methylated in this region. In addition, it was observed that all the methylated cell lines (Patu-8898T, HCT116, Caco-2, Hec-1-a, EGI-1, 23132/87, TFK-1, LNCaP, SNU-1, and SKOV-3) did not express ciRS-7 (Fig 3A, 3B). However, we also observed that a few unmethylated cell lines, DU145 and MDA-MB-321, did not express ciRS-7.

To further investigate the epigenetic mechanisms behind ciRS-7 expression, we performed ChIP-qPCR analyses (Fig 3C) for the histone modifications, H3K9Ac (active mark) and H3K27me3 (bivalent mark). These analyses were conducted on selected male cell lines with no ciRS-7 expression and no promoter methylation (DU145), ciRS-7 expression and no promoter methylation (A549 and H1650), as well as in two cell lines with no ciRS-7 expression and high promoter methylation levels (HCT116 and Caco-2). First, to validate the antibodies, we analyzed the H3K9Ac and H3K27me3 levels at the *UBC* and *RARβ1* promoter regions. As expected, we observed high levels of H3K9Ac and low levels of H3K27me3 at the *UBC* promoter and low levels of H3K9Ac and high levels of H3K27me3 at the *RARβ1* promoter (Fig 3D). Next, when analyzing the T3 promoter, we observed the highest H3K9Ac levels in A549, intermediate levels in DU145 and H1650, whereas it was almost absent in HCT116 and Caco-2, indicating that relatively high levels of H3K9Ac are needed for active transcription of the gene (Fig 3E). H3K27me3 was present in all the analyzed cell lines (Fig 3F), despite their differing ciRS-7 levels, indicating that this epigenetic modification is negligible for the epigenetic regulation of ciRS-7 in adenocarcinomas.

## Epigenetic silencing of ciRS-7 can be reversed in vitro

To further investigate the impact of epigenetic modifications on ciRS-7 expression, we treated selected cell lines with the demethylating agent, 5-azacytidine (5-aza) [32], and the HDACi, SAHA [33]. These experiments allowed us to assess whether DNA methylation and histone acetylation play a significant role in regulating ciRS-7 expression. First, we treated HCT116 cells, which harbored a fully methylated T3 promoter, with 5-aza alone. Upon treatment with 2.5 µM 5-aza, an increase in both ciRS-7 and T3 expression was observed (Fig 4A, 4B), accompanied by a decrease in methylation levels (Fig 4C). To determine if ciRS-7 expression could be increased in an unmethylated cell line, the prostate cancer cell line DU145 was treated with SAHA. This treatment also resulted in an increased ciRS-7 and T3 expression (Fig 4D, 4E), indicating that histone acetylation also plays an important role in the regulation of ciRS-7 expression. This was further investigated in HCT116, where ciRS-7 expression could be increased to much higher levels by combining 2.5 µM 5-aza with SAHA, compared to treatment with 5-aza alone (Fig 4F). In summary, demethylation and histone acetylation of the T3 promoter strongly activate ciRS-7 expression.

## Inverse correlation between ciRS-7 expression and T3 methylation in patient samples

Next, we investigated if DNA methylation levels also correlate with ciRS-7 expression in patient samples. To this end, FFPE colon cancer patient samples were laser-capture microdissected to separate stromal and cancer cell fractions, whilst expression of ciRS-7 and methylation levels of the T3 promoter were assessed. For each patient, higher ciRS-7 expression was observed in the stromal fractions (Fig 5A), while the methylation levels of T3 were higher in the cancer

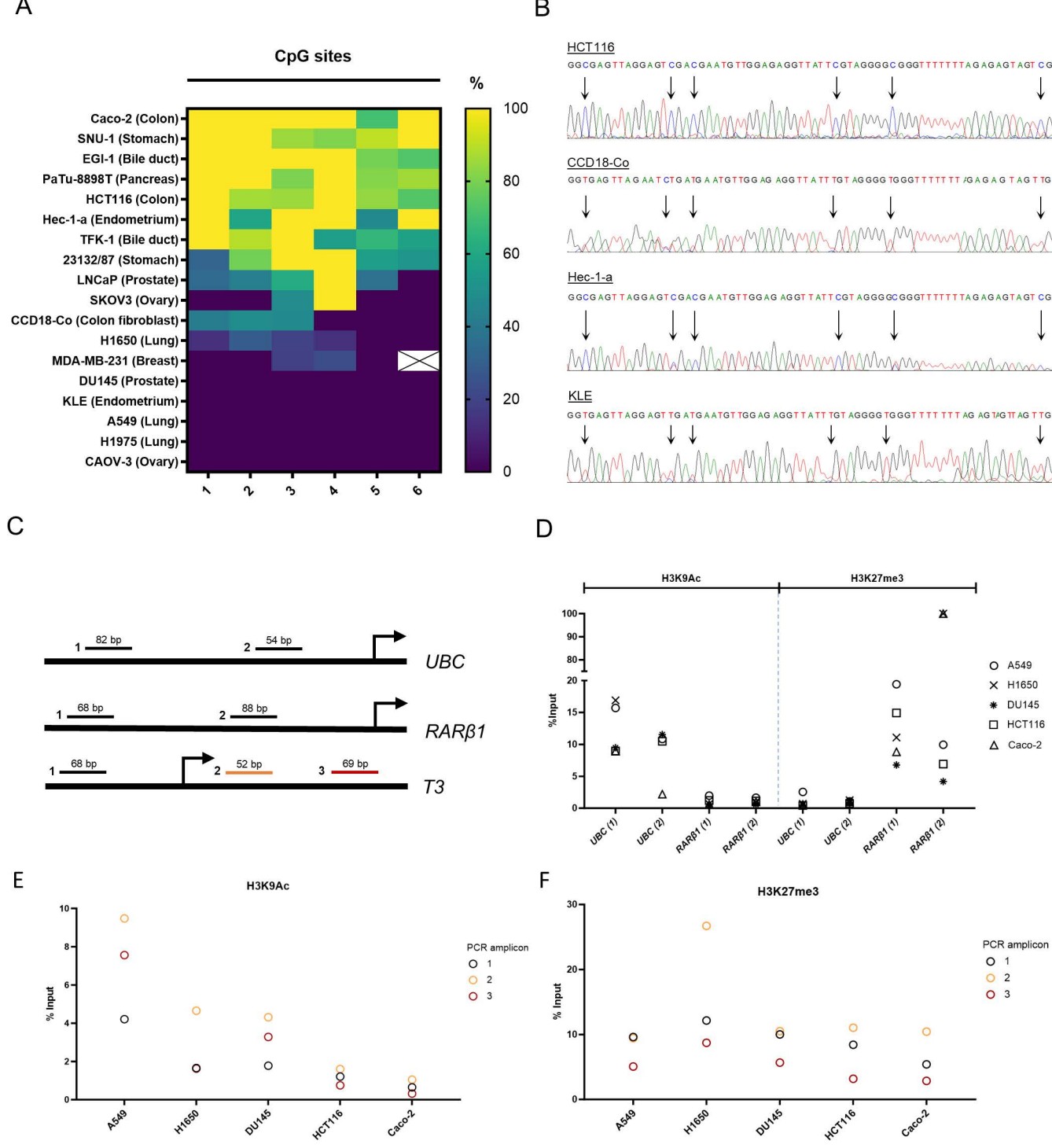

**Fig 3. ciRS-7 expression correlates with epigenetic silencing.** (A) Heatmap showing the percentage of methylation at the six investigated CpG sites in the T3 promoter across the cell lines. The percentage was calculated as described in the methods section. (B) Representative Sanger sequencing electropherograms of selected cell lines with various degrees of methylation. Arrows indicate CG sites in the sequence. (C) Illustration of the qPCR targets on *UBC*, *RARβ1*, and *T3* with transcription start site and qPCR amplicons indicated (not drawn to scale). Numbers indicate amplicons seen in (D, E, F). For *UBC* and *RARβ1* primers were designed targeting the genomic region 1000 basepairs (bp) upstream the transcription start site. (D) Chromatin immune-precipitation (ChIP) control primers for H3K9Ac (left) and H3K27me3 (right) for cell lines A549, DU145, HCT116, Caco-2, and H1650.

Enrichment was determined as % of input relative to the adjusted input control, calculated as $100 \cdot 2^{\Delta Ct}$. (E, F) Histone modifications, H3K9Ac (E) and H3K27me3 (F), at the T3 promoter region measured by ChIP in cell lines A459, DU145, HCT116, Caco-2, and H1650. Three sets of primers were used targeting three different regions of the *T3* promoter. Enrichment was determined as % of input relative to the adjusted input control, calculated as $100 \cdot 2^{\Delta Ct}$.

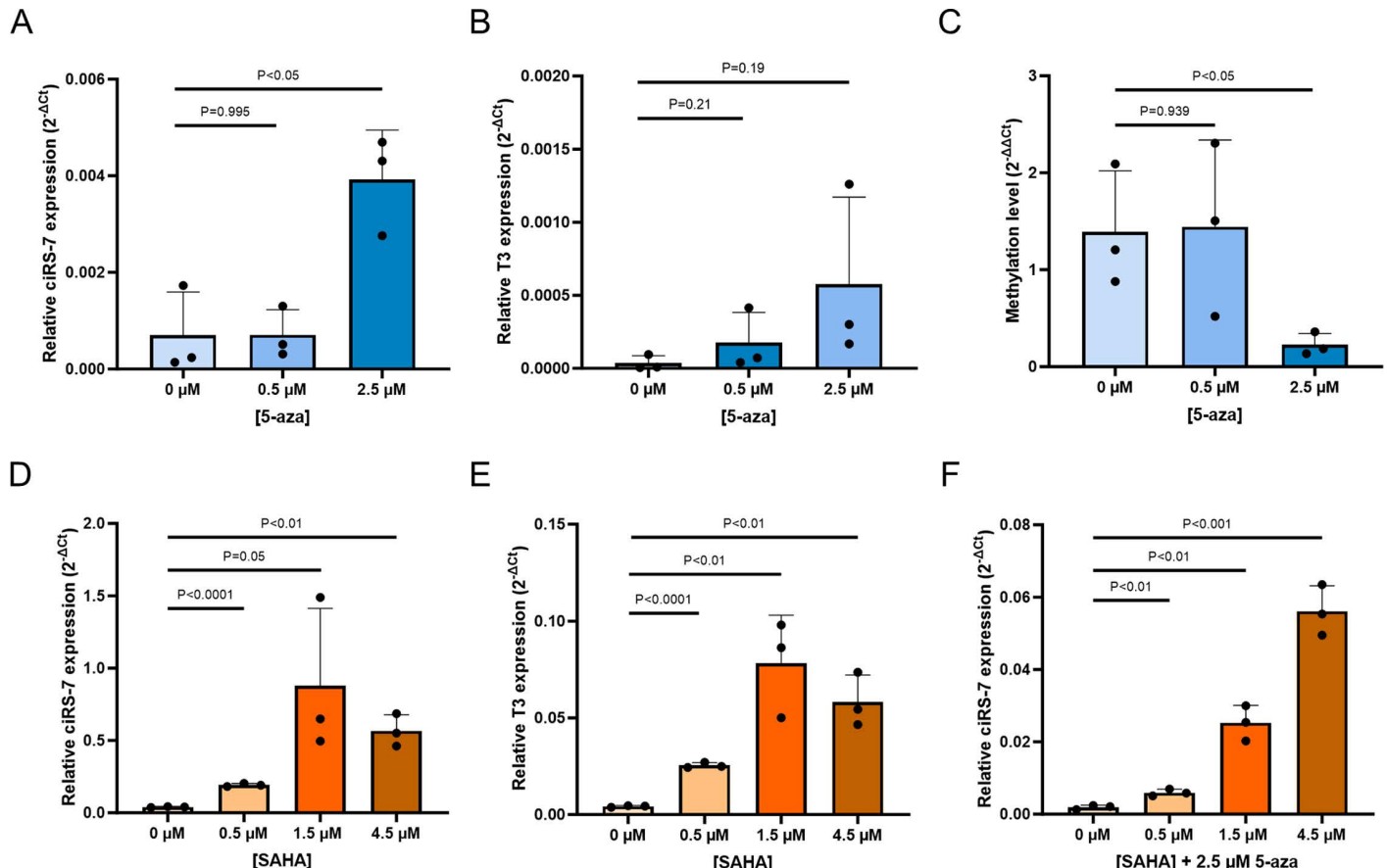

**Fig 4. ciRS-7 expression can be induced by reversing epigenetic modifications.** (A) ciRS-7 expression in HCT116 after treatment with 5-azacytidine (5-aza) measured by RT-qPCR. (B) T3 expression in HCT116 after treatment with 5-aza measured by RT-qPCR. (C) Methylation levels analyzed by Sensitive Melting Analysis after Real-Time Methylation-Specific PCR (SMART-MSP) of T3 in 5-aza treated HCT116 cells relative to methylated control DNA. (D) ciRS-7 expression measured by RT-qPCR in DU145 after treatment with Vorinostat (SAHA) at different concentrations. Dots represent biological replicates. (E) T3 expression measured by RT-qPCR in DU145 after treatment with SAHA at different concentrations. Dots represent biological replicates. (F) ciRS-7 expression in HCT116 after treatment with 2.5 μM 5-azacytidine and SAHA measured by RT-qPCR. Dot represents biological replicates. Unpaired T-tests were employed to calculate if changes in ciRS-7 expression, T3 expression, and T3 methylation after treatment were significant.

cells (Fig 5B). In summary, ciRS-7 expression and T3 methylation are inversely correlated in colon adenocarcinoma patient samples (Fig 5C).

## Discussion

To date, ciRS-7 is the most studied circRNA in cancer due to its numerous MREs for the tumor suppressor miR-7. However, many published studies are controversial because ciRS-7 may not be expressed in cancer cells while being abundant in the tumor microenvironment (TME) as observed in colon cancer [19,24]. Moreover, the epigenetic mechanisms

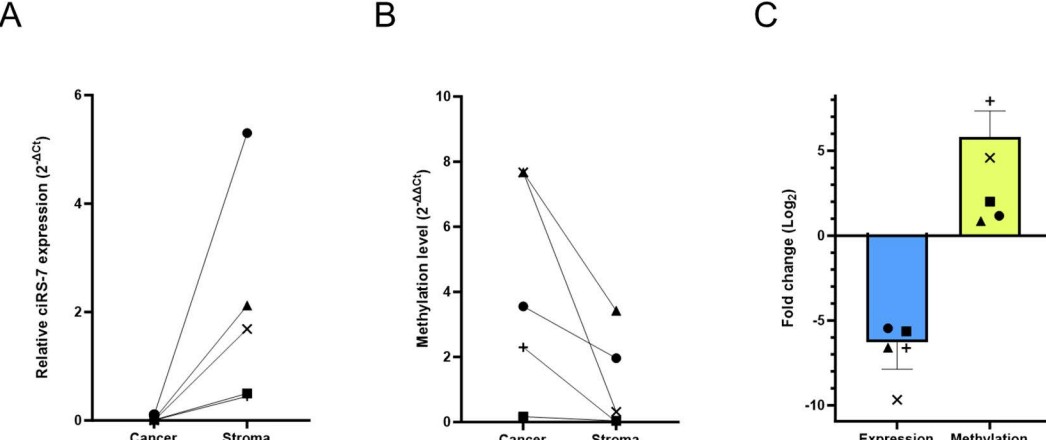

**Fig 5. ciRS-7 expression and T3 methylation are inversely correlated in cancer and stromal cells in colon adenocarcinomas.** (A) ciRS-7 expression in cancer and stromal cells of colon adenocarcinomas separated with laser-capture microdissection and quantified using RT-qPCR. Each symbol connected by a straight line reflects a primary tissue sample (n = 5). (B) Methylation level of the T3 promoter in cancer and stromal cells of laser-captured microdissected colon adenocarcinoma tissues measured by SMART-MSP. Each symbol connected by a straight line reflects a primary tissue sample (n = 5). (C) Bar-plot showing the fold change of ciRS-7 expression and T3 methylation between cancer and stromal cells in each patient sample. Each symbol reflects a primary tissue sample (n = 5).

behind ciRS-7 expression have not previously been examined in the context of adenocarcinomas. In this study, we examined the spatial expression patterns and epigenetic regulation of ciRS-7 across nine different types of adenocarcinomas using cell lines and primary patient samples. Using single molecule *in situ* hybridization of ciRS-7 in primary adenocarcinoma samples, we found that ciRS-7 was completely absent in the cancer cells in most cases, while being expressed in the stromal parts of the tumors. However, some variation across the adenocarcinoma subtypes was observed, in line with a previously published meta-analysis [34]. In particular, we found that ciRS-7 was expressed in the cancer cells within most breast cancer specimens.

When studying 18 cell lines representing nine different types of adenocarcinomas, we found that ciRS-7 expression correlates with the expression of the upstream T3 transcript. Interestingly, this transcript harbors a CpG island where methylation has previously been observed to correlate with ciRS-7 expression in multiple myeloma cell lines [16]. We demonstrated that H3K9 acetylation, but not H3K27 methylation, of the T3 promoter CpG island are important epigenetic modifications that impact ciRS-7 expression in adenocarcinomas, though H3K9Ac levels varies between cell lines that express ciRS-7. In the 18 studied adenocarcinoma cell lines, ciRS-7 was generally absent but was detected in two lung, one endometrial, and one ovarian cancer cell line. ciRS-7 has previously been shown to be highly expressed in neurons [18], and the high ciRS-7 expression previously observed in melanomas may be linked to the origin of melanocytes, which arise from the neural crest during embryonic development [35]. Moreover, high ciRS-7 expression has been observed in pancreatic islets containing a subset of neural cells [36]. Therefore, the observed expression of ciRS-7 in a few of the cell lines may potentially be attributed to neuroendocrine differentiation, which has been observed in some carcinomas of epithelial origin [37,38]. Indeed, the A549 lung cancer cell line, which harbored the highest levels of ciRS-7, has been described to undergo neuroendocrine differentiation [39].

The expression of ciRS-7 in cell lines was consistent with the absence of DNA methylation in the T3 promoter CpG island, and in laser-capture microdissected tissues, we observed an inverse correlation between ciRS-7 expression and T3 methylation levels. Although ciRS-7 expression was found to be regulated by promoter methylation, we observed that histone acetylation around the promoter also influences ciRS-7 expression. However, other regulatory mechanisms must be present for ciRS-7 to be expressed, as evidenced by our ChIP experiment where the prostate cancer cell line DU145

had intermediate levels of H3K9Ac despite not expressing ciRS-7. Nevertheless, when treating cell lines with a combination of 5-aza and SAHA, ciRS-7 expression increased greatly, while in the unmethylated cell line DU145, SAHA treatment alone was enough to increase ciRS-7 expression 10-fold.This study is the first to investigate the epigenetic regulatory mechanisms behind ciRS-7 expression in adenocarcinomas, highlighting the importance of understanding how the epigenetic control of ciRS-7 contributes to its proposed oncogenic functions in cancer. Due to the high intracellular stability of circRNAs, as well as their versatile regulatory functions, it is possible that ciRS-7 contributes to the regulation of a variety of genes related to TME function and development. Therefore, future studies of ciRS-7 and its oncogenic functions should focus on the TME. Indeed, ciRS-7 has been suggested to be important for intestinal epithelium homeostasis and repair [40], and it is possible that these functions are also important in TME development. Thus, ciRS-7 could be a potential target for future therapeutic strategies.

As ciRS-7 expression in the TME is activated during cancer progression [19], it is possible that changes in the epigenetic patterns of the cells within the TME may drive this phenomenon. Interestingly, the loss of the epigenetic factor SIN3B, which is involved in histone deacetylation, has been related to stromal activation of pancreatic adenocarcinomas [41,42].

Furthermore, ciRS-7 has potential as a diagnostic and prognostic biomarker in several cancers [21,43,44], and even as a non-invasive biomarker as exemplified in multiple myeloma and epithelial ovarian cancer [45,46].

In summary, our study provides key insights into the expression patterns and epigenetic regulation of ciRS-7 across all major adenocarcinomas. These findings are essential for guiding future research, which should focus on understanding the oncogenic potential of ciRS-7, which, based on our findings, appears to be driven by cells in the TME rather than by the cancer cells themselves.

## Materials and methods

### Ethics statement

The study was conducted according to the principles of the Declaration of Helsinki and approved by the Danish National Committee on Health Research Ethics (NVK) (2016120). Formal consent was waived as the study included retrospective samples of deceased patients.

**Patient samples.** Three tissue microarrays (TMAs) of 5 µm formalin-fixed and paraffin embedded (FFPE) adenocarcinomas were used in this study. One contained one core each representing breast, lung, prostate, pancreas and colon adenocarcinomas, while the remaining two TMAs contained cores originating from breast, lung, prostate, pancreatic, colon, bile duct, endometrial, and ovarian cancer, and three cores originating from stomach adenocarcinomas. Additionally, for stomach, bile duct, endometrial, and ovarian cancer, 5 µm thick FFPE full tissue sections were used. The total number of patient samples for each adenocarcinoma type was five for breast, lung, prostate, pancreas, colon, bile duct, endometrium, and ovary, and four for stomach. The tissue sections were cut at the pathology department at Aarhus University Hospital.

**Cell cultures.** Cell lines (Caco-2, HCT116, TFK-1, EGI-1, SKOV-3, CAOV3, Hec-1-a, KLE, SNU-1, 23132/87, A459, H1650, H1975, LNCaP, DU145, MDA-MB-231, and PaTu-8898T) representing nine different adenocarcinomas (colon, bile duct, ovarian, endometrial, stomach, lung, prostate, breast, and pancreatic) were grown under standard conditions, in accordance with the providers' recommendations at $37°C$ and 5% $CO_2$. The only exception was MDA-MB-231, which was cultured in 100% air. The normal colon fibroblast cell line, CCD18-Co, was cultivated in MEME (Sigma-Aldrich) containing 10% FBS (Sigma-Aldrich), 100 units/mL penicillin and 100 µg/mL streptomycin (Sigma-Aldrich), 2 mM L-glutamine (Sigma-Aldrich), 1mM sodium pyruvate (Sigma-Aldrich), and 1% non-essential amino acids (Sigma-Aldrich). Cell viability and cell counts were monitored using the Luna automated cell counter (Logos biosystems, Gyeonggi-do, South Korea)

Human cell lines were acquired from the American Tissue Culture Collection (ATCC, Manassas, VA, USA), Deutsche Sammlung von Mikroorganismen und Zellkulturen (DSMZ, Leibniz, Germany), or kindly donated by Karina Dalsgaard Sørensen and Lene Nejsum from Aarhus University Hospital (Aarhus, Denmark) or David Olagnier, Martin Roelsgaard

Jakobsen, and Naiara Santana Codina (Department of Biomedicine, Aarhus University). All cell lines were tested negative for mycoplasma and all donated cell lines were authenticated by Eurofins Genomics Germany Gmbh (Ebersberg, Germany)

**DNA and RNA purification.** Total DNA and RNA were isolated from the cell lines using the AllPrep DNA/RNA/miRNA Universal Kit (Qiagen), and total RNA was isolated using the QIAwave RNA Mini kit (Qiagen), both according to the manufacturer's protocol. DNA and RNA extraction from FFPE patient samples were performed using the Allprep DNA/RNA FFPE kit (Qiagen) according to manufacturer's protocol. The concentrations of total DNA and RNA were measured by spectrophotometry using a NanoDrop One spectrophotometer (Thermo Fisher).

**ciRS-7 chromogenic and fluorescent in situ hybridization.** The localization of ciRS-7 within primary FFPE tissue samples was investigated by single molecule chromogenic and fluorescent *in situ* hybridization using a slightly modified version of the BaseScope Detection Reagent Kit v2- RED protocol (Advanced Cell Diagnostics [ACD], MN, USA), where the nuclei of the cells in the tissues were stained with DAPI prior to mounting. The BaseScope method allows the detection of ciRS-7 using a BSJ-specific probe (BA-Hs-CDR1as-circRNA-Junc, ACD). In brief, the FFPE tissue samples were deparaffinized using Neoclear (Sigma-Aldrich) and treated with pretreatment buffers, allowing the probe to hybridize to ciRS-7 during the hybridization step. Eight amplification steps allowed the signal to be detected using FAST-Red, creating both a chromogenic and fluorescent signal. Counterstaining was done using 50% hematoxylin (Sigma-Aldrich) and 1 μg/ml DAPI (Thermo Fisher Scientific) dissolved in PBS. Data were collected using a Slide Scanner (Upright Widefield Fluorescence, Olympus VS120, Olympus Life Science, Japan) and analyzed using OliVIA (Ver. 2.9.1, Olympus Life Science) and QuPath (Ver. 0.3.1.) [47].

**RT-qPCR analyses of ciRS-7 and the LINC00632 transcripts T1, T2, and T3.** The M-MLV Reverse Transcriptase kit (Invitrogen, Waltham, MA, USA) was used to reverse-transcribe 500 ng of RNA according to the manufacturer's protocol and diluted 1:10 in nuclease free water (VWR, Avantor, PA, USA). RT-qPCR was performed using 4 μL of diluted cDNA and 6 μL of qPCR master mix consisting of 5 μL SYBR green PCR Master mix (Applied biosystems, Thermo Fisher Scientific, MA, USA) and 5 μM of each primer (S1 Table). The PCR amplification was performed using a LightCycler 480 instrument (Roche Life Science, Mannheim, Germany) with the following cycling conditions: one cycle of 95 °C for 10 min, followed by 45 cycles of 95 °C for 10 seconds, 60 °C for 30 seconds, and 72 °C for 30 seconds. The commonly used reference genes *GUSB*, *UBC*, and *PUM1* [48–50] were used to normalize the data using geometric means and relative gene expression was calculated as $2^{-\Delta Ct}$ [51]. Laser-capture microdissected samples were normalized to *UBC*. The experiments were done as technical triplicates at the qPCR level.

**NanoString nCounter analysis of ciRS-7 across cell lines.** The NanoString nCounter SPRINT instrument (NanoString Technologies, USA) was used to analyze 100 ng of total RNA using a custom panel (S2 Table) including ciRS-7, the T1, T2, and T3 transcripts according to the manufacturer's instructions with a 20-hour hybridization time. The data were processed with the nSOLVER 4.0 software (NanoString), and normalization was performed relative to the positive controls included in the panel, as well as to the geometric mean of the reference genes *ACTB, MRPL19, PUM1, SF3A1,* and *GUSB* [48]. The data were exported to Excel (Microsoft Corporation, USA) and the background threshold was defined as the mean of the negative controls plus two times the standard deviation.

**Bisulfite treatment and sequencing of the LINC00632 T3 promoter CpG island.** Bisulfite-treatment of 500 ng of genomic DNA for each sample was performed using the EpiTect Bisulfite Kit (Qiagen) according to the manufacturer's protocol. A PCR to amplify the T3 promoter was subsequently performed using Taq recombinant polymerase (Thermo Fisher) according to the manufacturer's recommendations with 10 mM dNTPs (Invitrogen, USA) and 10 μM forward and reverse primers from a previously published assay [16]. The PCR was run with the following cycling conditions: 94 °C for three minutes, 45 cycles of 94 °C for 20 seconds, 60 °C for 30 seconds, and 72 °C for 30 seconds followed by one cycle of 72 °C for 10 minutes. Subsequently, the amplicons were Sanger sequenced as previously described [16]. Sanger sequencing chromatograms can be found in S2 File. The methylation percentage of individual CpG sites was quantified

by assessing the peak intensities of thymine and cytosine using SnapGene Viewer (v. 6.1.2, GSL Biotech, CA, USA). Subsequently, the ratio between the bases was calculated, considering intensities under 100 as background [52].

**Laser-capture microdissections.** Five μm thick FFPE colon adenocarcinoma tissue sections from five colon cancer patients were mounted on membrane slides (Arcturus PEN Membrane Glass Slides, Applied Biosystems, USA) and stained with Mayers hematoxylin at the pathology department at Aarhus University Hospital (Aarhus, Denmark) and at Aarhus University. Laser-capture microdissection was performed on a Leica LMD 7 Laser Microdissection Microscope, where the cancer cell and stromal cell fractions were collected in 0.2 ml tubes respectively. Samples were stored at -18°C until they were processed as described above in the DNA and RNA extraction section.

**Sensitive Melting Analysis after Real-Time Methylation-Specific PCR (SMART-MSP).** SMART-MSP [53] was performed on bisulfite treated cell line DNA as described above. A previously published assay, which is less susceptible to normalization errors caused by copy number changes and aneuploidy, was used for normalization [54,55], while previously published primers were used to specifically target several CpG sites in the T3 promoter [16]. Bisulfite-converted fully methylated and fully unmethylated DNA (Epitect Control DNA, Qiagen) were used as positive and negative controls, respectively. The negative control was considered negative when amplification occurred after >35 PCR cycles. qPCR was performed using a 96-well plate with 2 μL of bisulfite-treated DNA and 8 μL of LightCycler 480 High-Resolution Melting Master (Roche Life Science) including primers, in each well as technical triplicates. The PCR amplification was carried out on a LightCycler 480 instrument II (Roche Life Science) with the following cycling conditions: one cycle of 95 °C for 10 min, followed by 45 cycles of 95 °C for 10 seconds, 60 °C for 20 seconds and 72 °C for 20 seconds. The melting program was carried out using the following conditions: 95 °C for 15 seconds, 40 °C for 1 minute, and 20 acquisitions/°C from 40 °C to 95 °C. The experiments were conducted using technical replicates at the level of qPCR on three biological replicates. Methylation levels were calculated using the $2^{-\Delta\Delta Ct}$ method [51] when comparing the samples to a fully methylated genomic control.

**5-aza and SAHA treatments.** In 6-well plates, 100,000 cells were seeded 24 hours prior to treatment. 5-aza monotreatment was performed every 24 hours for four days with either 0.5 μM or 2.5 μM 5-aza (STEMCELL technologies, Canada) freshly prepared in low FBS media and harvested 120 hours after seeding. Cells treated with media containing PBS instead of 5-aza served as a negative control. SAHA monotreatment was performed by seeding 100,000 cells in 6-well plates 24 hours prior to treatment, followed by treatment every 24 hours with either 0.5 μM, 1.5 μM, or 4.5 μM SAHA (MedChem Express LLC, USA) dissolved in DMSO for three days before harvesting after 96 hours. The negative control was treated with 0.3% DMSO. For 5-aza and SAHA combination treatment, 100,000 cells were seeded in 6-well plates 24 hours prior to treatment, followed by treatment with 2.5 μM 5-aza every 24 hours freshly prepared in low FBS media for four days. Forty-eight hours after seeding, the cells were treated with either 0.5 μM, 1.5 μM, or 4.5 μM SAHA dissolved in DMSO every 24 hours simultaneously with the 5-aza treatment for three days. The negative control was treated with PBS for the first 24 hours, and with PBS and 0.3% DMSO for the remaining timepoints. The cells were harvested after 120 hours, and RNA and DNA purifications were performed as previously described.

**Chromatin Immunoprecipitation (ChIP)-qPCR.** To study the presence of histone modifications, ChIP-qPCR was performed targeting H3K9Ac using anti-H3K9Ac (Thermo Fisher) or H3K27me using anti-H3K27me3 (Thermo Fisher) antibodies. ChIP was performed using the MAGnify Chromatin Immuniprecipitation System (Invitrogen, Waltham, MA, USA) as described by the manufacturer, using mouse IgG as a negative control. The antibodies were conjugated to Dynabeads Protein A/G for 1 hour at 4 °C. In brief, 1,000,000 cells were harvested and fixated with 37% formaldehyde (Sigma-Aldrich) for 10 minutes. The reaction was stopped using 1.25 M glycine. The cells were subsequently washed with cold PBS (VWR), pelleted (13,000 rpm at 4°C for 10 minutes), and lysed using lysis buffer. Ultrasonication was performed at high intensity for 15 minutes on the Bioruptor Sonication system Version 1.1 (Diogenode, USA) to create 200–500 basepairs (bp) long DNA-chromatin fragments, which were validated using gel electrophoresis. The chromatin was diluted 1:10 and conjugated antibodies were added. An input control was reserved for the reverse crosslinking and DNA purification steps. After IP, the DNA-chromatin complexes were reverse crosslinked, and DNA was purified using

DNA purification magnetic beads according to the MAGnify protocol. qPCR was performed as previously described using primers targeting the genomic region 1,000 bp upstream from the T3 transcript, whilst *UBC* and *RARβ1* were employed as controls for H3K9Ac and H3K27me3, respectively (S3 Table). The Ct values for the T3 transcript were normalized to the 1% input control, which was adjusted to 100% by subtracting 6.6 cycles from the measured Ct value, while the degree of histone modification (% Input) was calculated as $100 \cdot 2^{\Delta Ct}$.

**Statistical analysis.** To assess potential linear correlations, linear regression with an F-test for significance was employed. Unpaired *t*-tests were performed to compute the *P*-values between groups as detailed in the figure legends. Plotting of data and statistical analyses were performed using Prism 10.0.2 (GraphPad, USA). *P*-values below 0.05 were considered statistically significant.

## Supporting information

**S1 Fig. ciRS-7 expression is present in the cancer cells in some primary adenocarcinoma tissue samples.** A-I) Chromogenic *in situ* hybridization of ciRS-7 in bile duct (A), pancreatic (B), prostate (C), lung (D, E), and breast (F-I) adenocarcinomas. ciRS-7 signals are observed as pink dots. Scale bars are indicated in the lower-left corners (A = 100 μm, B-I = 50 μm).
(TIF)

**S1 File. Pictures of all patient samples analyzed using FISH.**
(ZIP)

**S2 File. Sanger sequencing chromatograms.**
(ZIP)

**S1 Table. qPCR-primer list.**
(PDF)

**S2 Table. nCounter gene panel.**
(PDF)

**S3 Table. ChIP-PCR primers.**
(PDF)

**S4 Table. Underlying numerical data for all graphs.**
(XLSX)

## Acknowledgments

The authors would like to thank Professor Martin Roelsgaard Jakobsen, Associate Professor David Olagnier, Professor Karina Dalsgaard Sørensen, Assistant Professor Naiara Santana Codina, and Professor Lene Nejsum at Aarhus University for providing cell lines used in this study. We also thank the staff at Aarhus University Hospital for the preparation of patient samples and laboratory technician Kim-Gwendolyn Dietrich and biomedical laboratory scientist Mariana Semenova for technical assistance. We also thank the Imaging Core Facility of the Department of Biomedicine at Aarhus University for use of imaging equipment.

## Author contributions

**Conceptualization:** Thea P. Paasch, Morten T. Jarlstad Olesen, Jørgen Kjems, Henrik Hager, Lasse S. Kristensen.
**Data curation:** Thea P. Paasch, Adrienne M. Assmus.

**Formal analysis:** Thea P. Paasch, Juan L. García-Rodríguez.

**Funding acquisition:** Robert A. Fenton, Lasse S. Kristensen.

**Investigation:** Thea P. Paasch, Henrik Hager.

**Project administration:** Thea P. Paasch.

**Resources:** Morten T. Jarlstad Olesen, Robert A. Fenton, Henrik Hager.

**Supervision:** Lasse S. Kristensen.

**Visualization:** Thea P. Paasch.

**Writing – original draft:** Thea P. Paasch, Lasse S. Kristensen.

**Writing – review & editing:** Thea P. Paasch, Morten T. Jarlstad Olesen, Juan L. García-Rodríguez, Adrienne M. Assmus, Robert A. Fenton, Jørgen Kjems, Henrik Hager, Lasse S. Kristensen.

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
