## [Decision Letter · Decision Letter 0]

4 Mar 2025

PGENETICS-D-24-01216

ciRS-7 expression is epigenetically regulated in cancer cells across human adenocarcinomas

PLOS Genetics

Dear Dr. Sommer Kristensen,

Thank you for submitting your manuscript to PLOS Genetics. After careful consideration, we feel that it has merit but does not fully meet PLOS Genetics's publication criteria as it currently stands. Therefore, we invite you to submit a revised version of the manuscript that addresses the points raised during the review process.

Please submit your revised manuscript within 60 days May 03 2025 11:59PM. If you will need more time than this to complete your revisions, please reply to this message or contact the journal office at plosgenetics@plos.org. Please include the following items when submitting your revised manuscript:

We look forward to receiving your revised manuscript.

Kind regards,

Michael J Metzger, Ph.D.

Academic Editor

PLOS Genetics

Kent Hunter

Section Editor

PLOS Genetics

Aimée Dudley

Editor-in-Chief

PLOS Genetics

Anne Goriely

Editor-in-Chief

PLOS Genetics

**Additional Editor Comments:**

This manuscript has been reviewed by three reviewers (I apologize for the delay in getting reviewers), and all reviewers note interest in the findings and also noted significant issues. We thank the authors for this manuscript, and we will consider a revised manuscript once the reviewers' comments have been addressed. Of note, multiple reviewers brought up the issue that there may only be a single sample for each tissue type and a small number of cell lines tested, and heterogeneity could be a concern in the generalizability of the findings, so this issue should be addressed clearly.

**Journal Requirements:**

1) We do not publish any copyright or trademark symbols that usually accompany proprietary names, eg ©,  ®, or TM  (e.g. next to drug or reagent names). Therefore please remove all instances of trademark/copyright symbols throughout the text, including:

- ® on pages: 14, 17, and 18

- TM on pages: 13, 14, and 18.

2) In the online submission form, you indicated that data presented in this manuscript is available upon request.  All PLOS journals now require all data underlying the findings described in their manuscript to be freely available to other researchers, either

1. In a public repository

2. Within the manuscript itself

3. Uploaded as supplementary information.

3) Please amend your detailed Financial Disclosure statement. This is published with the article. It must therefore be completed in full sentences and contain the exact wording you wish to be published.

**Reviewers' comments:**

Reviewer's Responses to Questions

Reviewer #1: The study first analyzed ciRS-7 expression levels and spatial distribution in nine types of adenocarcinomas and 18 cell lines, revealing a positive correlation between ciRS-7 expression and the T3 transcript of LINC00632. Further investigations demonstrated that DNA methylation and H3K9 acetylation were critical epigenetic modifications regulating ciRS-7 (LINC00632) expression, while H3K27 methylation showed no significant impact. Through inhibitor experiments, the regulatory roles of H3K9 acetylation and DNA methylation on ciRS-7 expression were confirmed. Additionally, using laser-capture microdissection to distinguish cancer cells from the tumor microenvironment (TME), the study established that ciRS-7 expression was predominantly localized in the TME. However, I have some questions regarding the research content:

1. Fig. 1: How many samples were used for tissue FISH validation of ciRS-7 in each type of adenocarcinoma? If only one sample was analyzed for each cancer type, it might not be sufficient to conclude that ciRS-7 is not expressed in a particular cancer.

2. Although LINC00632 is the host gene of ciRS-7 and positively correlated with ciRS-7 expression, Fig. 2F indicates that H3K27me3 does not show significant differences in LINC00632 (T3). Thus, the statement in Line 172–173, "H3K27me3 was present in all cell lines (Fig. 3F) despite the large differences in ciRS-7 expression," is not accurate and needs revision.

3. The FFPE colon cancer patient samples in Fig. 5 include only three samples, which raises concerns about potential sampling bias.

4. The key finding of this study was that the transcription of LINC00632 was regulated by epigenetic modifications. As the host gene of ciRS-7, LINC00632 may have influenced ciRS-7 expression. Therefore, ciRS-7 itself may not have been directly regulated by histone modifications.

Spelling error:

Line 217: "adenocarcinoma patient samples (Fig. 5. C)" need corrected to " ...(Fig. 5C)".

Reviewer #2: In this study, the Authors investigated the expression and epigenetic regulation of the circular RNA ciRS-7 across 9 types of adenocarcinomas. They demonstrate that ciRS-7 is absent in cancer cells but expressed in the tumor microenvironment (TME). Moreover, epigenetic modifications, specifically DNA methylation and H3K9 acetylation, were found to regulate ciRS-7 expression. This is an interesting and thorough study that provides insights into the role of ciRS-7 in cancer biology and its potential as a therapeutic target. However, there are several issues in the manuscript:

1. While the presence of ciRS-7 in the TME is well-described, the manuscript lacks experimental evidence on its specific functional roles. Additional experiments or cited literature linking ciRS-7 in the TME to tumor progression, immune modulation, or therapeutic resistance would significantly strengthen the conclusions of the study.

2. The Methods section should include more details, for instance regarding normalization strategies for RT-qPCR and nCounter data, and sample size limitations for certain experiments.

3. The absence of ciRS-7 in cancer cells is demonstrated. Did the Authors address variability across cancer subtypes within adenocarcinomas? Is there any data regarding heterogeneity in ciRS-7 regulation across tumor stages or grades?

4. The Discussion could include more in-depth commentary. For example, the potential oncogenic role of ciRS-7 in TME is briefly mentioned, but the Authors do not explore therapeutic targeting strategies. Discussion on targeting epigenetic modifiers or disrupting ciRS-7 interactions would provide translational relevance.

5. It would be interesting to also discuss the potential role of ciRS-7 as a non-invasive biomarker. The Author could refer to recent studies that suggest its diagnostic/prognostic role in cancer, such as: Papatsirou et al., EJHaem. 2024;5(4):677-689.

Reviewer #3: In this manuscript, Paasch and coauthors characterize the expression of ciRS-7 in different tumor contexts. While absent in most adenocarcinoma cells, it was found in the tumor microenvironment. Additionally, the authors investigate the epigenetic mechanisms that regulate ciRS-7 expression.

The specific localization of ciRS-7 within the microenvironment rather than in cancer cells was unexpected, providing valuable insights into the relevant milieu for circRNA function. The study is well-conducted and clearly presented, although a more in-depth exploration of ciRS-7's role specifically in the stroma would be beneficial.

Major points

1. The correlation between T3 and ciRS-7 expression is based on only a few cell lines (Figure 2). To further substantiate the link between these two RNAs, T3 transcript expression should also be analyzed in Figure 4 following 5-aza and SAHA treatment. Similarly, are T1-T3 also differentially expressed between adenocarcinoma cells and the stroma (Figure 1)?

Minor points

1. Line 165: Should read “…as well as in two cell lines with no ciRS-7 expression and high...”

2. Line 170: Typo correction: "A459" should be "A549."

3. Figure 3E: Only two replicates are shown for the H1650 cell line. For robustness, please include a third replicate.

4. Figure 4: To better interpret the extent of reactivation upon treatment with different epigenetic modulators, expression levels should be normalized to baseline (conc=0).

5. Lines 262-263: Please review and revise the syntax for clarity.

**Have all data underlying the figures and results presented in the manuscript been provided?**

Reviewer #1: None

Reviewer #2: Yes

Reviewer #3: Yes

PLOS authors have the option to publish the peer review history of their article (what does this mean? ). If published, this will include your full peer review and any attached files.

**Do you want your identity to be public for this peer review?** For information about this choice, including consent withdrawal, please see our Privacy Policy .

Reviewer #1: No

Reviewer #2: **Yes: ** Christos K. Kontos

Reviewer #3: No

**Figure resubmission:**
---

## [Decision Letter · Decision Letter 1]

13 May 2025

Dear Dr Sommer Kristensen,

We are pleased to inform you that your manuscript entitled "ciRS-7 expression is epigenetically regulated in cancer cells across human adenocarcinomas" has been editorially accepted for publication in PLOS Genetics. Congratulations!

Yours sincerely,

Michael J Metzger, Ph.D.

Academic Editor

PLOS Genetics

Kent Hunter

Section Editor

PLOS Genetics

Aimée Dudley

Editor-in-Chief

PLOS Genetics

Anne Goriely

Editor-in-Chief

PLOS Genetics

Comments from the reviewers (if applicable):

The authors have satisfied all three reviewers' questions and concerns in the revised manuscript, and we will like to inform you that we will be accepting it for publication. I apologize for the delay and thank you for working and publishing with us.

Reviewer's Responses to Questions

**Comments to the Authors:**

Reviewer #1: The authors have addressed all my concerns in the revised manscript.

Reviewer #2: The Authors have adequately addressed the Reviewers’ comments; the appropriate corrections were made and incorporated into the text and the revised manuscript is significantly improved. The clarifications that are provided contribute to the coherence and quality of the study. Overall, the paper is well-written and contributes to the existing knowledge in its field and the Reviewer suggests that it is suitable for publication.

Reviewer #3: In their revised manuscript, the authors have addressed my previous issues. I am satisfied with the updated manuscript and recommend it for publication.

**Have all data underlying the figures and results presented in the manuscript been provided?**

Reviewer #1: Yes

Reviewer #2: Yes

Reviewer #3: Yes

PLOS authors have the option to publish the peer review history of their article (what does this mean? ). If published, this will include your full peer review and any attached files.

**Do you want your identity to be public for this peer review?** For information about this choice, including consent withdrawal, please see our Privacy Policy .

Reviewer #1: No

Reviewer #2: **Yes: ** Christos K. Kontos

Reviewer #3: No

**Data Deposition**

http://datadryad.org/submit?journalID=pgenetics&manu=PGENETICS-D-24-01216R1

**Press Queries**

---

## [Editor Report · Acceptance letter]

PGENETICS-D-24-01216R1

ciRS-7 expression is epigenetically regulated in cancer cells across human adenocarcinomas

Dear Dr Sommer Kristensen,

We are pleased to inform you that your manuscript entitled "ciRS-7 expression is epigenetically regulated in cancer cells across human adenocarcinomas" has been formally accepted for publication in PLOS Genetics! Your manuscript is now with our production department and you will be notified of the publication date in due course.

With kind regards,

Zsofia Freund

PLOS Genetics

On behalf of:
